# Properties of 3D-Printed Polymer Fiber-Reinforced Mortars: A Review

**DOI:** 10.3390/polym14071315

**Published:** 2022-03-24

**Authors:** Jie Liu, Chun Lv

**Affiliations:** 1College of Light-Industry and Textile Engineering, Qiqihar University, Qiqihar 161006, China; 01250@qqhru.edu.cn; 2Engineering Research Center for Hemp and Product in Cold Region of Ministry of Education, Qiqihar University, Qiqihar 161006, China; 3College of Architecture and Civil Engineering, Qiqihar University, Qiqihar 161006, China

**Keywords:** polymer fiber reinforcement, 3D printing, geopolymer, mortar, working performance

## Abstract

The engineering applications and related research of fiber-reinforced cement and geopolymer mortar composites are becoming more and more extensive. These reinforced fibers include not only traditional steel fibers and carbon fibers, but also synthetic polymer fibers and natural polymer fibers. Polymer fiber has good mechanical properties, good bonding performance with cement and geopolymer mortars, and excellent performance of cracking resistance and reinforcement. In this paper, representative organic synthetic polymer fibers, such as polypropylene, polyethylene and polyvinyl alcohol, are selected to explore their effects on the flow properties, thixotropic properties and printing time interval of fresh 3D-printed cement and geopolymer mortars. At the same time, the influence of mechanical properties, such as the compressive strength, flexural strength and interlaminar bonding strength of 3D-printed cement and geopolymer mortars after hardening, is also analyzed. Finally, the effect of polymer fiber on the anisotropy of 3D-printed mortars is summarized briefly. The existing problems of 3D-printed cement and polymer mortars are summarized, and the development trend of polymer fiber reinforced 3D-printed mortars is prospected.

## 1. Introduction

In recent years, the engineering applications and related researches of fiber-reinforced cement-based composites and geopolymer composites have become increasingly widespread [1,2,3]. These reinforced fibers not only include traditional fibers, but also synthetic polymer fibers [4,5,6,7] and natural polymer fibers [8,9]. At the same time, these fibers are also used in 3D-printed mortars (3DPMs) [10,11,12]. The core principle of 3D-printed technology is layered additive manufacturing, including extrusion stacking process, powder process, sliding form process and other construction processes [13,14]. At present, 3D-printed technologies in the field of architecture mainly adopt D-Shape technology and Contour crafting [15,16]. D-shape technology is a powder construction process. Contour crafting is an extruded stacking process, in which only the external contour of the component is printed, and then mortar is filled into the contour [17,18]. 3DPM technology does not need templates, and has the advantages of fine process and rapid molding. Its material properties include working performance, mechanical property, and durability [19,20]. 3DPMs with a continuous uniform extrusion can be the first and second layers as well as the adhesiveness between the layers and the requirement of setting time; free forming can be built to meet mortar performance [21,22,23]. Due to technologies, 3DPMs are generally unable to install a reinforcing steel bar, so the most commonly used method is to add fiber material, in order to enhance the structural strength, toughness and impact resistance [24,25]. Some scientists suggest adding prestressed reinforcement after mortar forming and hardening, but the process is complicated [26].

For more than a decade, the researches and applications of polymers as 3D-printed building mortar materials have attracted more and more attention from scientists. As green building materials, geopolymers can replace cement-based materials as 3D-printed building mortars in the future [27,28,29]. In this paper, the properties of the polymer fiber-reinforced 3D-printed building mortars used in recent years are summarized. The mortar types discussed mainly include geopolymer mortars and cement mortars. The effects of different types of polymer fibers on the working performances and mechanical properties of 3DPMs are summarized, and the interlayer bonding strength and anisotropy of mortars are analyzed.

Traditional fibers used in 3DPMs include steel, carbon, glass, and basalt. Steel fibers can effectively improve the tensile and bending ability of mortar composites, and have impact resistance and fatigue resistance. The disadvantage of steel fibers is that they are easy to rust and the price is relatively high [30]. Carbon fibers have the characteristics of being lightweight and having high strength, high temperature resistance, corrosion resistance, and fatigue resistance, but they also have poor toughness and insufficient impact resistance. Glass fibers have good corrosion resistance and heat insulation, but their alkali and wear resistance are poor [31,32,33]. Traditional fibers can reduce the brittle failure of 3DPMs, but strain softening occurs when mortar is strained [34]. Replacing traditional fibers with organic synthetic polymer fibers can effectively reduce costs and improve the performance of 3DPMs [35]. Polymer fiber is a kind of synthetic monofilament bundle high strength fiber, and adding it into 3DPMs can effectively control the formation and development of cracks, improve its impact resistance and impermeability. Commonly used adulterated polymer fibers include polyethylene (PE) [36], polypropylene (PP) [37], polyvinyl alcohol (PVA) [38], polyacrylonitrile (PAN) [39], aramid (AR), and polyphenylene benzodioxazole (PBO).

## 2. Polymer Fibers

The mechanical properties, geometric characteristics and elongation of polymer fibers can affect the properties of 3DPMs. A comparison of the typical performance values of commonly used polymer fibers is shown in Table 1 [40]. It can be seen that PVA and AR are both high elastic modulus fibers, which not only have mechanical properties similar to steel fibers, but also have excellent fatigue resistance and impact resistance. The other polymer fibers have a low elastic modulus, which mainly improve the toughness and impact resistance of 3DPMs.

The 3DPM mixture includes cementing material, aggregate, admixture, fly ash, silica powder, other admixtures and an appropriate water–binder ratio. To improve mortar properties, researchers also added functional materials, such as fibers. As fiber type and dosage, fiber length and aspect ratio affect mortar properties, there are significant differences in mortar properties with different types of polymer fibers. This is closely related to the fiber’s own performance. On the other hand, the different bond characteristics between fibers and matrix also affect the macroscopic mechanical properties of mortars. Table 2 shows fiber types and the material composition of mortar mixture used in 3DPM experimental study. The cementitious material in the table is 100. Other materials are converted according to the percentage of cementitious material, and the input of cement and other materials is averaged.

It can be seen that the volume fraction of the polymer fiber content of 3DPMs is generally not more than 2.0%, and the polymer fibers with an aspect ratio ranging from 30 to 150 are more suitable to be added to 3DPMs. Currently, studies on the impact of polymer fibers are mainly focused on PVA, PP, PE, etc., while there are relatively few studies on the impact of other polymer fibers on the properties of 3DPMs.

## 3. Effect of Fiber on 3DPM Workability

The workability of ordinary mortars mainly refers to working performance, which is the most important performance of mortar mixture, mainly including fluidity, cohesion and water retention. Due to the requirements of the mortar printing process, in addition to fluidity, it is necessary to control its thixotropy and setting time, which is the key to affect whether mortar can be continuously and uniformly extruded and ensure pouring molding. In addition to selecting suitable cementitious material, fine aggregate, admixture and other necessary materials, adding fibers into mortar can restrain the deformation of the mortar structure, prevent the spalling of printing mortar, and obtain a uniform and continuous printing effect. When printing mortar, fiber content has a certain influence on the sensitivity of nozzle blockage. Compared to steel fibers, flexible polymer fibers have less influence [19].

### 3.1. Influence of the PE Fiber

In the polymer fiber mixed with printing mortars, PE fiber is widely used. PE fiber is a strong lightweight thermoplastic polymer material with a high strength and low density. Ye et al. conducted an experimental study on the influence of fluidity of 3DPMs under different PE fiber contents [44]. It was found that the expansion diameter and penetration depth of mortar decrease with the increase in fiber content, and the fluidity became worse. The expanded diameter of printing mortar with 2.0% fiber content was 144 mm, which was 11.1% and 14.8% lower than that of mortar with 1.5% and 1.0% PE fiber content, respectively. Similarly, the penetration depth of printed mortar with 2.0% fiber content was 76 mm, while the penetration depth of mortar with 1.5% fiber content was 80 mm and that with 1.0% fiber content was 86 mm, respectively, improving by 5.0% and 11.6%. The experimental comparison between PE fiber of different lengths and other types of polymer fibers and steel fibers showed that both steel fiber and polymer fiber mortars had the properties of controlling cracking and increasing toughness after loading to the peak load. The experimental results also showed that, different from steel fibers, the fluidity of mortar mixed with polymer fibers was worse, but the sudden fracture of mortar mixed with steel fibers was avoided at the late loading stage, and the application performance of polymer fibers was higher than that of steel fibers [43].

### 3.2. Influence of PP Fiber

Among the polymer fibers used in 3DPMs, there are many cases of PP fiber. Researchers adjusted the working performance of mortar by adding 0.25%, 0.5%, 0.75% and 1.0% PP fibers [41]. The mix proportions of 3D-printed geopolymer mortars is shown in Table 3. The experimental results showed that all samples had reasonable machinability and extrusion performance. Through the fluidity test, the change of the expanded diameter of the mixture with different PP fiber content is shown in Table 3. Similar to PE fibers, increasing the content of PP fiber also decreases the fluidity of mortar.

The working performance of mortar was improved by adding carboxymethyl cellulose (CMC) into mortar mixture. On the one hand, increasing the content of PP fiber reduces the fluidity of the mixture; on the other hand, adding CMC improves the fluidity of the mixture. However, the experiments showed that reducing the fiber content had a greater impact than increasing the content of CMC. Table 3 shows that the PP fiber mortar mixture with 0.75% content exhibits the highest extension diameter of 158 mm, and this abnormal behavior requires further mechanism study.

### 3.3. Influence of PVA Fiber

When PVA fiber is added into 3DPMs, the fluidity of mortar decreases obviously with the increase in fiber content. Studies showed that the fluidity of polymer fiber 3DPMs should be controlled within 170 mm to 180 mm, and the PVA fiber content of 1% to 1.4% was in line with this range [47,48]. Under the premise of the same amount of retarder, the influence range of mortar setting time was less than 6 min with the increase in the amount of PVA fibers, indicating that the amount of PVA fibers had no obvious influence on the mortar setting time.

Malaszkiewicz et al. [49] selected three polymer fibers with a length of 12 mm to 50 mm and a volume fraction of 0% to 4% to conduct mortar fluidity and setting time experiments. According to the test results, it was found that fiber type and fiber length had no significant effect on the yield strength of mortars. As time goes on, the loss of liquidity slowed down compared with plain mortar. Le et al. [50] developed 3DPMs with a setting time of 10 min to 80 min and a maximum of 100 min, all of which had a good mortar printing quality. Thixotropic property is the key to ensure vertical stacking and rapid prototyping of 3DPMs. The rheometer was used to test the thixotropic performance of 3DPMs. It was found that the thixotropic performance index reached 10,000 N·mm·rp, and the mortar material could be better printed and exported. Experiments showed that adding polymer fibers could effectively improve the thixotropic properties of 3DPMs [25].

In fact, different types of polymer fibers have little effect on the performance of printed mortar mixture. The fluidity of the mixture decreased when different polymer fibers were added. The decrease in the fluidity of mortar mixture has no negative impact on 3DPMs, because the printing of mortar materials needs to have high thixotropy and constructability at the same time, and the addition of polymer fibers can make the mortar mixture achieve a better shape retention.

## 4. Effect of Fiber on Mechanical Properties of Mortars

Mechanical properties of mortars mainly include compressive strength, tensile strength, flexural strength, flexural strength and shear strength. 3DPMs also include the interface bonding strength between adjacent mortar printing layers, as well as the anisotropic effect caused by different printing layers and different orientations of fibers in mortars.

### 4.1. Influence of Single Polymer Fiber

Similar to single traditional fiber reinforced mortars [51], single polymer fibers, such as PE [43,44,52], PP [24,41,42,43] and PVA [25,32,43,45], have a great influence on the mechanical properties of 3DPMs. PE fiber reinforced mortar has excellent properties. In China, Tsinghua University used 3DPM to build a 26.3 m long, 3.6 m wide single-arch structure pedestrian bridge, using PE fiber mortar composites, with a compressive strength up to 65 MPa and flexural strength up to 15.6 MPa [52]. The researchers carried out a series of uniaxial tensile, compression and bending tests using PE fibers. The results showed that the tensile strength and strain capacity of the mortar with a fiber content of 1.5% and 2.0% were slightly lower than that of the experimental group without fiber, while the tensile strain capacity of mortar with fiber content of 1.0% was slightly increased. The bending strength, deformation capacity and energy dissipation capacity of 1.5% sample were better than those of other samples. Significant anisotropy was observed in compressive strength and flexural energy dissipation capacity. However, the observed anisotropy is negligible [44].

The effect of PP fibers on the mechanical properties of 3D-printed geopolymers is similar to that of cement-based materials. The relationship between the vertical bending stress and mid-span deflection of printed mortar was studied by adding 0.25% to 1.00% PP fibers into a 3D-printed geopolymer mixture. It was found that the mixture without fiber (PP_0_) showed brittle failure, and the load was reduced to zero after the sample was broken. The failure mode of the fiber-doped material changes from brittleness to toughness, and its properties depend on the fiber content. Fiber content 0.25% (PP0.25) and fiber content 0.5% (PP0.5) showed flexural softening characteristics, while fiber content 1% (PP1.00) and 0.75% (PP0.75) showed flexural hardening behavior [41].

PVA fibers were used to prepare 3DPMs, and the penetration strength was 0.05 MPa at the initial stage of printing, 0.7 MPa after 30 min, and 40 MPa after 28 days [25]. Experiments on PVA fibers show that they can effectively improve the compressive strength of mortars, and the compressive strength of 1 day can reach 50% of the compressive strength of 28 days, which is suitable for 3DPM rapid construction requirements. With the increase in PVA fiber content, the flexural strength of mortar increases obviously. PVA fiber plays an effective bridging role in the early hydration process of mortar, and the bridging effect is obvious to improve the bending strength of the mortar. Especially in the process of mortar printing, the PVA in the mortar is oriented along the printing path, so as to significantly improve the bending strength of mortar.

Scanning electron microscopy test was conducted on mortar specimens damaged by compression [53]. Figure 1a,b is a scanning electron microscopic image of PVA after breaking. PVA is obviously pulled out from mortar, and micro-scratches are formed on the fiber surface. It can be seen that the fracture of PVA is in the shape of sheet tear which indicates that the fiber bridging the surface of the cracks in fiber-reinforced geopolymer mortar consume energy when it is pulled out. The tests show that the bonding performance between PVA fibers and mortar is good, which can effectively prevent the development of micro-cracks in the mortar. Meanwhile, the microstructure analysis of Figure 1c,d also shows that the PE fiber rearranges inside the mixture during the printing process [44]. As the fiber content increased to 1.5% and 2.0%, more and more fibers developed twisted orientation. When the specimen is fractured under tensile stress, the tensile load is completely dependent on the bridging capacity of the fibers. If the fibers are twisted together, the stress between the mortar interfaces cannot be directly transferred to the fibers.

### 4.2. Influence of the Hybrid Polymer Fibers

Based on the advantages and disadvantages of single fiber mortars, many scientists have used two or more kinds of fibers with different properties and characteristics to prepare hybrid fiber mortars to meet the needs of different engineering. The hybrid of polymer fibers and traditional fibers into 3DPMs can give full play to the respective properties of fiber and further improve the mechanical properties of 3DPMs.

Through the mortar bending experiment of PP and PVA fibers, the effect of hybrid fibers on mechanical properties of mortar is similar to that of single fibers. With the increase in the proportion of PP and PVA fibers, the plastic deformation capacity of mortar is further improved, and the number of cracks in mortar is significantly reduced [54].

There are many experiments on the mechanical properties of multi-fiber reinforced 3DPMs. The mechanical properties of mortar can be improved effectively by mixing polymer fibers and inorganic fibers. For example, PP fibers and alkali resistant glass fibers are selected to improve the surface crack resistance of mortar [44,50]. Some scientists use a mixture of fibers of different sizes. By mixing long steel wire and short PVA fibers, the flexural strength of the printed sample was increased by 290% [55]. Similar to the single fiber, the compressive strength does not change obviously after the addition of mixed fiber, but the tensile strength and flexural strength of mortar increase obviously when the fiber content reaches 1% volume content. By reducing the water–binder ratio and optimizing the material gradation, some researchers achieved a compressive strength of 57 MPa, a maximum of 100 MPa, and a flexural strength of 11 MPa, with good working performance [55].

Figure 2 summarizes the relationship between polymer fiber content and compressive strength of 3DPMs. It can be seen that the influence of different types of polymer fiber on the compressive strength of 3DPMs has no obvious difference. The fiber content selected in experimental study cases is mostly about 1%. Due to the restriction of fibers on mortar matrix, the compressive strength of 3DPMs increases with the increase in fiber content.

## 5. Influence of Fiber on Interlaminar Bonding Strength

Due to the layered printing, 3DPM properties have great performance differences in different directions of space, with obvious anisotropy and low compressive strength and elastic modulus in a certain direction [56,57,58,59]. On the one hand, macro-scale pores exist on the mortar interface, and the bond ability between structural layers is weak. On the other hand, during the flow process of mortar printing slurry, fibers tend to separate and float, resulting in a different orientation and distribution of fibers in the structure. Fibers are distributed in a directional direction parallel to the printing direction, that is, parallel to the interface between the printing layers (Figure 3).

Through testing the 3DPM prepared by PVA anti-cracking fibers, the interlayer bonding strength can reach 2.53 MPa [52]. PE, PP and other fibers have a certain negative impact on the interlayer bonding strength of 3DPMs [60,61]. Compared to the sample without fiber, the tensile strength and shear strength of the interlayer mortar with fiber did not increase, and even decreased. The main reason is that there is no fiber connection between the interfaces of the printing mortar layers, which leads to the failure of the fibers between the interfaces of the printing mortar layers. On the other hand, the fluidity of mortar is reduced after fibers are added, and the bond between the printing layer interfaces is relatively weakened, which further reduces the binding effect between the printing layers.

Figure 4 shows the compressive strength of mortar mixture in different directions. It can be seen from Figure 4 that the compressive strength of PP_0_ mortar in the vertical direction is about 22 MPa. Due to the addition of 0.25% PP fibers, the strength value of PP_0.25_ mortar is increased to about 36 MPa. This indicates that a large number of fibers are aligned parallel to the direction of extrusion, which contributes to crack bridging under vertical pressure. However, a further increase in fiber content results in a decrease in vertical compressive strength. This may be due to the fiber-induced increase in air, which increases the porosity of the mixture [62].

The printing interval between mortar layers, the height of printer nozzle and the dehydration of the mortar surface will affect the compressive strength and tensile strength of 3DPMs as well as the interlayer bonding strength. If the printing interval between adjacent layers of mortar is short, the effect of layer orientation is very limited. With the increase in the printing time interval between adjacent mortar layers, the bond strength between mortar layers will decrease. The height of printer nozzle has limited influence on the bonding strength between mortar layers [63]. There is a correlation between the bonding strength between mortar layers and the mortar setting time and the water content of the interface between mortar layers [64]. The water content of interlayer interface depends on the bleeding rate and surface water drying rate of mortar. The results also show that the anisotropy of mortar is more obvious in compressive strength than in flexural strength. Figure 5 shows flexural properties of geopolymer composite with 6 mm fiber length [33]. T1, T2 and T3 are different loading directions. T1 loading direction is perpendicular to the printing layer, while T2 loading direction is parallel to the printing direction. The loading direction of T3 is perpendicular to the printing direction. It can be seen that the deformation is minimum when the loading direction is perpendicular to the printing layer. In contrast, when the loading direction is perpendicular to the printing direction, the bending strength value is the lowest. Composite mortar has distinct anisotropy.

In view of the weakness of mortar interlayer bonding strength, many optimization methods have been proposed. When preparing mortar, the material that can enhance the bond property of the interface between mortar layers is added. It is also possible to add a layer of material with good bonding strength to the mortar material between the mortar layers [65,66,67,68,69], or use some nanomaterials to improve the bonding performance between the fibers and the mortar matrix [70,71,72,73,74,75,76]. In addition, the chemical modification of polymer fibers can be used to improve the construction process and increase the performance of printing equipment, so that the fiber can penetrate the interface between mortar layers and participate in the bonding force between mortar layers [77,78,79,80]. Through the improvement of materials, machinery and process factors, the reinforcement skeleton is added in the process of mortar printing, in order to achieve fiber reinforcement, reinforcement skeleton setting, prestressed tendon tension and other multi-channel structural reinforcement system [81,82,83,84].

The durability of mortar has an important effect on the long-term service life of the cement matrix structure. By studying the action mechanism of polymer fibers in mortars, it is found that at the initial stage of mortar cracking, the internal crack of fiber bridge restricts the cracking process of the printed mortar body and restrains the further development of cracks to a certain extent. It is found that when natural fibers, such as cellulose fibers, are used as reinforcement fiber, the geopolymer matrix is more beneficial to the durability of the composite compared with cement matrix. This is because the strong alkalinity of cement-based materials can dissolve and erode cellulose fibers [85,86,87].

In fact, compared to the research on the influence of polymer fibers on the working and mechanical properties of mortar, there are relatively few studies on the durability of polymer fiber-reinforced 3DPMs. The reason is that 3DPMs has only been developed in recent years, and there is still a lack of relevant polymer fiber reinforcement verification cases in engineering practice. Due to the short application time in natural environment, long-term relevant experimental research and engineering application are needed to accumulate relevant experience data and the verification of relevant mechanism in durability research.

## 6. Conclusions and Outlooks

At present, polymer fibers mostly use PE, PP and PVA, other polymer fiber application cases are few. Polymer fibers reduce the fluidity of mortar mixtures and achieve a better shape retention. There is little difference in the influence of polymer fibers on the compressive strength of mortar, and the flexural strength is obviously enhanced, but attention should be paid to the anisotropy of mortar and the orientation of fiber direction [88]. The fluidity of materials, printing direction and printing time have a significant influence on the overall bearing capacity of 3DPMs. Based on the domestic and foreign research work on the performance of polymer fiber reinforced 3DPM material, there are still problems to be faced in the next research works:Theoretical research on the interaction between polymer fibers and matrix of 3D mortar materials is insufficient. According to the characteristics of polymer fibers, the study of different types of polymer fiber theory support is insufficient, although the normal fiber reinforced mortar fiber spacing theory has been studied. 3DPM has its particularity, especially between the layers, and needs further research.There is a lack of engineering practice and application cases of polymer fiber reinforced 3DPMs, and the application of fibers in relevant experimental studies is relatively simple.The lack of chemical modification treatments of polymer fibers to improve its adhesion with matrix.The interaction mechanism of polymer fibers with nanomaterials, carboxymethyl cellulose and other additives needs further study.

The problems existing in the study on the influence of polymer fibers on the properties of 3DPMs should be further explored. The following are put forward for the future studies dealing with this subject:Replacing traditional steel fibers with polymer fibers can reduce the cost and improve the performance of 3DPMs. However, polymer fibers can easily pollute the environment during their production and use, which violates the concept of green environmental protection. It is imperative to replace polymer fibers and traditional fibers with natural plant fibers. Natural fibers are renewable and degradable, rich in resources, and can develop vigorous natural plant fiber-reinforced 3DPMs [89,90].Further study of the influence mechanism of anisotropy of 3DPMs, adjust the printing direction, fiber type and dosage according to the structural design, so that the anisotropy of mortar is consistent with the stress of the building.Hybrid fibers can improve the compressive strength, flexural strength and splitting strength, and significantly enhance the cracking resistance and toughness of 3DPMs. The hybrid fibers should be further combined with the characteristics of all kinds of fiber, using the principle of combination of different lengths, rigid and flexible, and actively promoting the use of hybrid fibers. The polymer fiber is soft and the steel fiber is hard, which can be mixed to avoid single fiber length and form a complete fiber network with reasonable fiber length gradation.The surface modification of polymer fibers can be combined with low temperature plasma, surface oxidation, surface grafting and silane coupling agent to improve the properties of fiber reinforced materials.The gap between each layer of mortar will affect the strength and durability of mortar. Through the rational use of fiber characteristics, the construction process conditions can be improved and the performance of printing equipment can be increased, so that the fiber penetrates the interface between mortar layers, takes part in the combination of mortar layers, and enhances the vertical and horizontal bonding of mortar layers.In the future, it is necessary to make full use of the effective combination of polymer fibers and nanomaterials to study the durability of 3DPMs. Using the flexibility of polymer fibers and large specific surface area of nanomaterials, multi-scale macro and micro fibers are formed in mortars, which further improve the workability, mechanical properties and durability of 3DPMs.Actively explore ways to improve the properties of 3DPMs. For example, it has been proven that it is possible to efficiently reinforce concrete beams by spatial 3D-printed polymer elements [91].

## Figures and Tables

**Figure 1 polymers-14-01315-f001:**
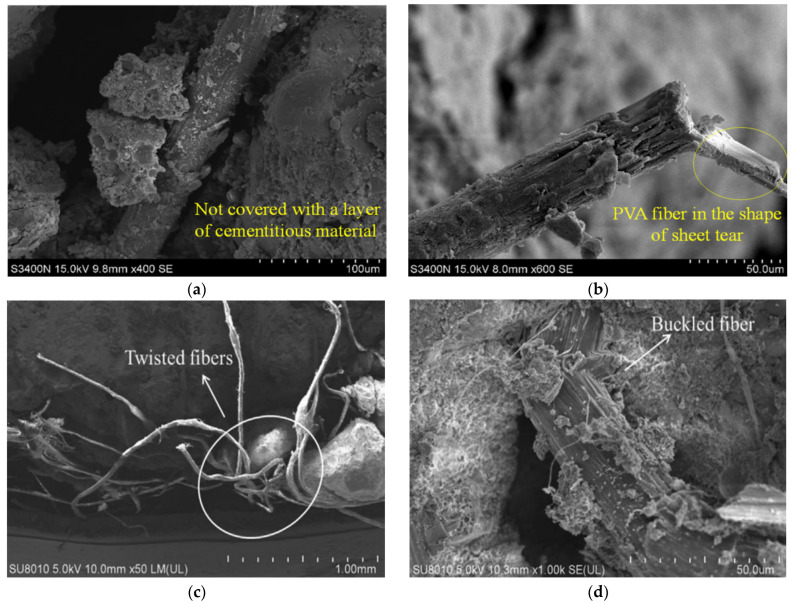
(**a**) Overlapping of fiber and matrix and (**b**) fiber pulling out from matrix. Adapted from Xiao et al. [53]; licensed under CC BY 4.0 (open access). SEM images of the (**c**) twisted fibers and (**d**) buckled fiber. Adapted with permission from [44], copyright 2021 Elsevier.

**Figure 2 polymers-14-01315-f002:**
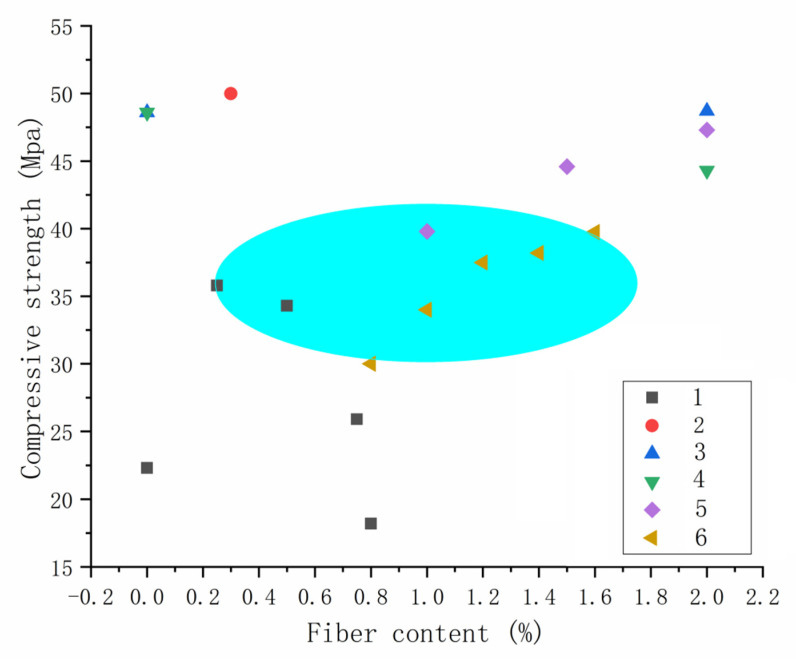
The relationship between the fiber content and compressive strength of the mortar. (In Figure 2, 1–6 are the relationship between fiber content and compressive strength in the cited literature. 1 is PP fiber [41], 2 is PVA fiber [46], 3 is PVA fiber [43], 4 is PE fiber [43], 5 is PE fiber [44] and 6 is PVA fiber [45]).

**Figure 3 polymers-14-01315-f003:**
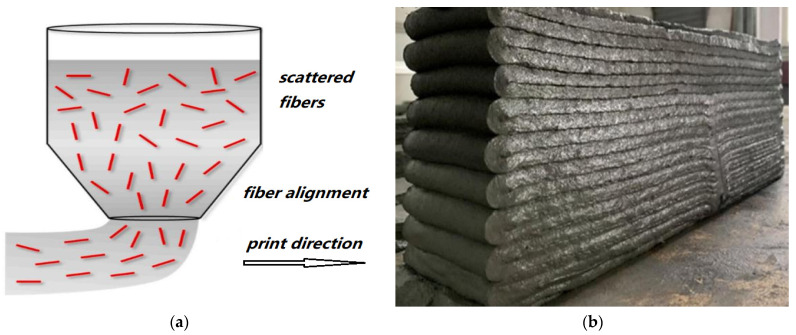
(**a**) Schematic drawing showing the admixture with fibers. Adapted from Nematollahi et al. [41]; licensed under CC BY 4.0 (open access). (**b**) Adhesion between the printing layers. Adapted with permission from [44], copyright 2021 Elsevier.

**Figure 4 polymers-14-01315-f004:**
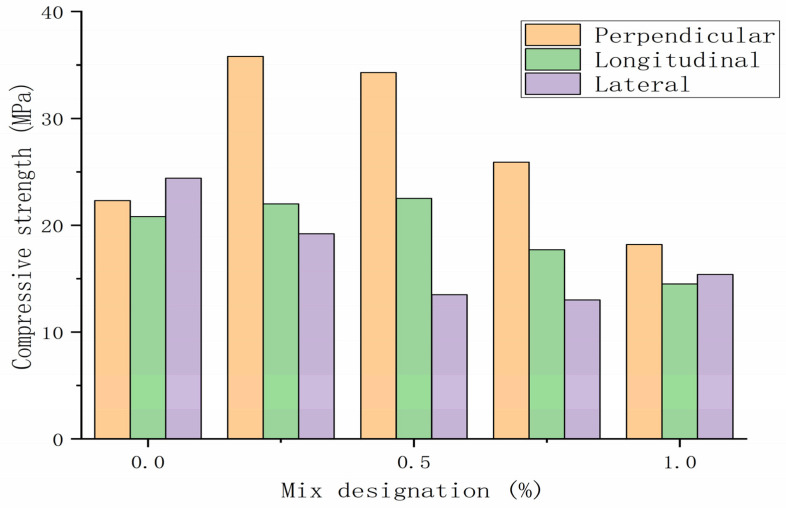
The compressive strength of printed specimens in different directions. Adapted with permission from [62], copyright 2019 Elsevier.

**Figure 5 polymers-14-01315-f005:**
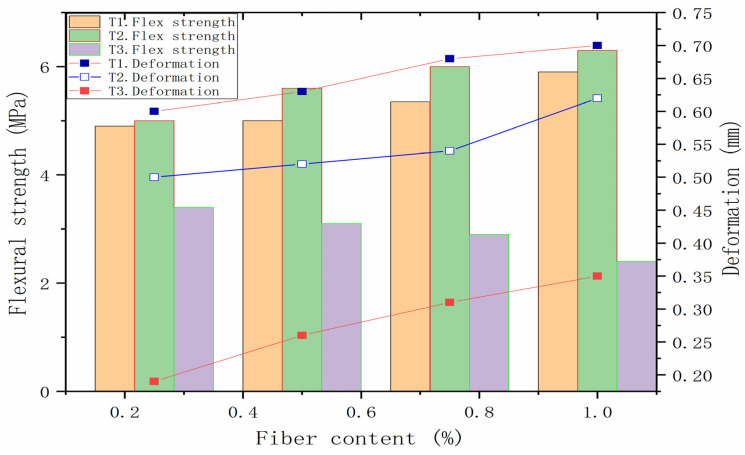
Flexural properties of the geopolymer with 6 mm fiber length. Adapted with permission from [33], copyright 2017 Elsevier.

**Table 1 polymers-14-01315-t001:** Properties of polymer fibers.

Fiber	Diameter (µm)	Density (g·cm^−3^)	Tensile Strength (MPa)	Young’s Modulus (GPa)	Ultimate Elongation (%)
PE	7–150	0.96	200–300	5–6	3–4
PP	20–70	0.91	300–700	3.5–11	15–25
PVA	1.30	1.30	600–2500	5–50	6–17
PAN	5–17	1.18	800–950	16–32	9–11
AR	10–12	1.442	500–3100	60–120	2.1–4.5

**Table 2 polymers-14-01315-t002:** Composition of the mortars.

Fiber Type	Fiber Length (mm)	Aspect Ratio	Fiber Content (%) *	Gelled Material	Admixture (%) *	Ref.
PP	12.00	667	1.20	OPC	20 (FA); 10 (SF)	[24]
	11.20	1867	0.25–1.00	GC	100 (FA)	[41]
	6.00	500	0.25	GC	100 (FA)	[42]
PE	12.00	1000	2.00	GC	50 (FA); 50 (Slag)	[43]
	12.00	480	1.00; 1.50; 2.00	OPC	17.99 (FA); 37.5 (SF); 8.69 (NR)	[44]
PVA	26.00	2167	0.25	GC	100 (FA)	[42]
	8.00	200	2.00	GC	50 (FA); 50 (Slag)	[43]
	9.00	290	0.8–1.6	OPC	50 (GGBS); 17 (SF)	[45]
	12.00	480	0.30	GC	11.12 (SAC)	[46]
PBO	11.20	933	0.25	GC	100 (FA)	[42]

* The fiber is given as volume content; other materials are given in mass content. OPC—ordinary Portland cement; GC—geopolymer; FA—fly ash; SF—silica powder; GGBS—mineral powder; SAC—sulphoaluminate cement.

**Table 3 polymers-14-01315-t003:** The mix proportions of 3D-printed geopolymer mortars.

Mix ID	Fly Ash	Activator	Fine Sand	Coarse Sand	PP Fiber	CMC	Expanded Diameter (mm)
PP0	1.0	0.467	1.135	0.365	-	0.040	147
PP0.25	1.0	0.467	1.135	0.365	0.25	0.028	139
PP0.5	1.0	0.467	1.135	0.365	0.50	0.023	138
PP0.75	1.0	0.467	1.135	0.365	0.75	0.012	158
PP1.00	1.0	0.467	1.135	0.365	1.00	0.004	134

## Data Availability

Not applicable.

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
