# Peer review of "Properties of 3D-Printed Polymer Fiber-Reinforced Mortars: A Review"

_polymers, 2022, doi:10.3390/polym14071315_

Round 1

Reviewer 1 Report

Dear Authors

Your article is very interesting and well organized.

However, it needs further improvement in order to be a better version. My suggestions are given in the attached copy of the article.

Please, try to use uniformly fibers instead of fiber, or abbreviations where they are suitable, etc...

I have commented on some of your subtitles, please check them.

Thank you.

Author Response

        Thank you very much for reviewing our manuscript in your busy schedule. Based on your valuable suggestions, we have carefully revised our manuscript. The goal point of this manuscript is achieved successfully. 

  1. it needs further improvement in order to be a better version. My suggestions are given in the attached copy of the article.

        We have made further improvements to the article as suggested by the reviewers. The revised content of the article has been marked in red.

  1. Please, try to use uniformly fibers instead of fiber, or abbreviations where they are suitable, etc...

Relevant content has been corrected in the manuscript according to the comments of reviewer.

  1. I have commented on some of your subtitles, please check them.

        We have corrected it according to the subtitles commented by the reviewer.

         (The revised page 3, lines 94-96; page 4, lines 141; page 5, lines 182-184; page 5, lines 196-197; page 5, lines 205-210; page 6, lines 231-234; page 7, lines 244-245; page 7, lines 267-269).  

        At the same time, some references have also been added. 

        (The revised page 15, lines 594-601).

Reviewer 2 Report

This research is very interesting, it provides very useful informations about the 3D printing of polymer fiber-reinforced mortars.

Although the paper is very well organized, with ideas and discussions well exposed, the following minor comments might be considered for improving the quality of the work.

- Most of the citations from the references do not comply with the format requested by the journal (pages 12-15). Please review this. For example, cometimes, there is the DOI and sometimes not. Same thing for the pages numbers.

- Fig. 2 and 3 are not easily readable (too small). Please correct.

- The figures aspects should be homogenized : authors should replot the figures 2, 4 and 5with the original results given in references in the same format. Please also check the references cited in Fig. 2 match with the ref cited in this article.

- You should consider this recent paper, dealing on the 3D printed  geometry of mortars : 10.3390/MA13143133

- Authors should propose some outlooks based on the conclusion for the future studies dealing of this subject.

Author Response

        Thank you very much for reviewing our manuscript in your busy schedule. Based on your valuable suggestions, we have carefully revised our manuscript. The goal point of this manuscript is achieved successfully. 

  1. Most of the citations from the references do not comply with the format requested by the journal (pages 12-15). Please review this. For example, cometimes, there is the DOI and sometimes not. Same thing for the pages numbers.

        Thank you very much for your review. According to your suggestion, we have carefully revised the manuscript. The references have been carefully corrected as suggested by the reviewers.

(The revised page 12-15, lines 413-616).

  1. Fig. 2 and 3 are not easily readable (too small). Please correct.

        Thank you very much for your review. According to your suggestion, we have carefully revised the manuscript. Figure 2 and Figure 3 have been corrected.

(The revised page 7, lines 255, page 8, lines 272).

  1. The figures aspects should be homogenized : authors should replot the figures 2, 4 and 5with the original results given in references in the same format.

        Thank you very much for your review. According to your suggestion, we have carefully revised the manuscript. Figure 2, Figure 4, and Figure 5 have been redrawn. The references cited in Figure 2 have been corrected.

(The revised page 7, lines 254, page 8, lines 272, page 9, lines 309, 311).

  1. Please also check the references cited in Fig. 2 match with the ref cited in this article.

        Thank you very much for your review. According to your suggestion, we have carefully revised the manuscript. The reference [43] cited in FIG. 2 has been corrected.

(The revised page 13, lines 506-508)

  1. You should consider this recent paper, dealing on the 3D printed  geometry of mortars : 10.3390/MA13143133

        Thank you very much for your review. According to your suggestion, we have carefully revised the manuscript. At the same time, some references have also been added.

(The revised page 15, lines 594-601).

  1. Authors should propose some outlooks based on the conclusion for the future studies dealing of this subject.

        Thank you very much for your review. According to your suggestion, we have carefully revised the manuscript. Authors have proposed some outlooks based on the conclusion for the future studies dealing of this subject.  

(The revised page 11, lines 369-371, page 12, lines 395-402).

Round 2

Reviewer 1 Report

Dear Authors

Your work is well structured and improved from the previous version. Some minor suggestions are given in the attached copy.

Thank you in advance for considering them and improving your manuscript.

Author Response

Thank you very much for reviewing our manuscript in your busy schedule. Based on your valuable suggestions, we have carefully revised our manuscript.

  1. The keyword “Mechanical property”has been removed. (The revised page 1, lines 23).
  2. The punctuation “.” has been removed.(The revised page 2, lines 70).
  3. Figure 2has been modified. (The revised page 7, lines 253).

Thank you very much for your review. According to your suggestion, we have carefully revised this part of the content.

Finally, thank you again for your wonderful review of our article in your busy schedule.
